# Directed Network Comparison Using Motifs

**DOI:** 10.3390/e26020128

**Published:** 2024-01-31

**Authors:** Chenwei Xie, Qiao Ke, Haoyu Chen, Chuang Liu, Xiu-Xiu Zhan

**Affiliations:** 1Research Center for Complexity Sciences, Hangzhou Normal University, Hangzhou 311121, China; 2College of Media and International Culture, Zhejiang University, Hangzhou 310027, China

**Keywords:** network comparison, motifs, Jensen–Shannon divergence, directed networks

## Abstract

Analyzing and characterizing the differences between networks is a fundamental and challenging problem in network science. Most previous network comparison methods that rely on topological properties have been restricted to measuring differences between two undirected networks. However, many networks, such as biological networks, social networks, and transportation networks, exhibit inherent directionality and higher-order attributes that should not be ignored when comparing networks. Therefore, we propose a motif-based directed network comparison method that captures local, global, and higher-order differences between two directed networks. Specifically, we first construct a motif distribution vector for each node, which captures the information of a node’s involvement in different directed motifs. Then, the dissimilarity between two directed networks is defined on the basis of a matrix, which is composed of the motif distribution vector of every node and the Jensen–Shannon divergence. The performance of our method is evaluated via the comparison of six real directed networks with their null models, as well as their perturbed networks based on edge perturbation. Our method is superior to the state-of-the-art baselines and is robust with different parameter settings.

## 1. Introduction

Many systems in various domains featuring intricate interaction relationships can be effectively represented in the form of complex networks [1], including social platforms [2,3], biological systems [4], and economic systems [5]. Due to the diversity of network forms [6,7] and the high-order features of networks [8,9], the precise measurement of similarity between different networks, namely, the design of an effective network comparison method, has emerged as a central focus in the field of network science. Network comparison aims to quantify the differences between two networks based on the network topological structure, allowing the effective handling of different types of tasks [10,11]. For example, in the field of pattern recognition, network comparison can be applied to classify contents such as images, documents, and videos [12]. In the biological domain, network comparison can be used to analyze which protein interactions may have equivalent functions [13]. In neuroscience, the comparison of brain networks contributes to understanding the functional differences between normal and pathological brains [14].

The original term used to compare networks was the graph isomorphism problem [15], which has been proven to fall within the NP complexity class [16]. In recent years, researchers have proposed various methodologies from different perspectives and technologies to measure the similarity between networks [17,18,19,20,21,22]. The majority of these methods have primarily concentrated on the comparison of undirected networks. However, interactions among distinct entities in the real world commonly exhibit asymmetry. In social networks, an instance of user *i* trusting user *j* does not necessarily imply reciprocal trust from *j* to *i*. The directionality of the interactions between nodes in a network, which can not be captured by an undirected network, has boosted the research of directed network comparison. For example, Bagrow and Bollt [23] utilized portrait divergence, a metric based on the distribution of the shortest path lengths, to evaluate the structural similarities between networks. Koutra et al. [24] proposed DeltaCon by calculating the Matusita distance of similarity matrices between two networks. Sarajlic et al. [25] extended network distance measures to directed networks using directed graphlets and demonstrated their efficacy in distinguishing various directed networks. Centrality-based methods, such as the degree [26], closeness [27], and the clustering coefficient [28], compare networks based on the centrality values of each node. Although these methods are capable of comparing networks effectively to some extent, most of them do not consider the higher-order structure, i.e., interactions among more than two nodes, of a network, which has been shown to be ubiquitous in various complex systems [9]. The interactions among multiple nodes have been modeled by simplexes, hypergraphs, and subgraphs such as motifs in work from different domains [29,30,31]. To capture the higher-order interactions between nodes in directed networks, we propose using direct motifs to quantify the dissimilarity between two networks. Motifs refer to recurring subgraphs in a network, where these subgraphs exhibit specific interaction patterns that facilitate the understanding of the functionality of networks [32]. Motifs have been widely used in different network tasks, i.e., community detection [33], link prediction [34], and node-ranking problems [35]. In contrast to traditional conventional methods, motif-based approaches consistently exhibit superior performance in tackling these problems.

To explore the similarity between different directed network structures, in this paper, we propose a motif-based directed network comparison method, $D_{m}$, i.e., using motifs to examine smaller components of directed networks to assess the similarity between networks. We start by constructing a node motif distribution matrix, where the elements in the matrix are obtained by computing the distribution of nodes appearing in different directed motifs. Due to computational complexity, we consider the motifs composed of 2 to 4 nodes and thus obtain 35 different directed motifs. Later on, we use the Jensen–Shannon divergence to quantify the dissimilarity between two directed networks both locally and globally. We validate the effectiveness of $D_{m}$ in six real directed networks. Compared to the baseline methods, $D_{m}$ exhibits notable distinguishability and robustness in comparing networks.

The rest of this paper is organized as follows. Section 2 introduces the definition of motifs in a directed network and details the motif-based directed network comparison method. We provide a clear description of the baseline methods and directed network datasets in Section 3. All experimental results are presented in Section 4. Section 5 summarizes the full paper.

## 2. Method

### 2.1. The Definition of Motifs in a Directed Network

A directed unweighted network is represented by G=(V,E), where $V=\left\{v_{1},v_{2},\cdots ,v_{N}\right\}$ and $E=\left\{e_{k}=\left(v_{i},v_{j}\right)\left|k=1,\cdots ,M\right|v_{i},v_{j}\in V\right\}$ are the sets of nodes and edges, respectively. The numbers of nodes and edges are given by *N* and *M*. The adjacent relationship between two nodes in *G* is given by the adjacency matrix *A*, with $A_{ij}=1$ indicating that there is a directed edge between $v_{i}$ and $v_{j}$ and $A_{ij}=0$ implying that there are no edges between them. We note that the directionality of *G* determines that *A* is an asymmetric matrix.

Motifs are the most common graphical patterns in complex networks, consisting of a group of closely connected nodes and edges. Due to the high complexity of computing motifs in a network, we normally consider motifs formed by 2 to 4 nodes. Motifs play a crucial role in the study of complex networks, acting as fundamental building blocks for large complex networks, analogous to genes in biology. In a directed network, the motifs are formed by nodes with directed edges. We show examples of directed motifs in Figure 1. There are 35 directed motifs, each comprising 2 to 4 nodes, individually represented by $m_{1}$ to $m_{35}$, respectively. For instance, there are two kinds of motifs if we consider two nodes, which are given by $m_{1}$ and $m_{2}$ in the figure.

### 2.2. The Motif-Based Directed Network Comparison Method

Motifs contain important topological information of a network and thus are essential for network comparison. Based on the distinctive topological properties of directed motifs, we first compute the motif distribution in a directed network. As the time complexity of computing motifs is quite high, we will use the motifs listed in Figure 1 that are formed by 2,3, and 4 nodes for the computation of motif distribution. Specifically, we use $T_{i}=\left\{t_{i}\left(j\right)|1\leq j\leq 35\right\}$ to represent the motif distribution of node $v_{i}$, where $t_{i}\left(j\right)$ represents the fraction of motif *j* that contains $v_{i}$. Consequently, an N×35 matrix $T=\left\{T_{1},T_{2},\cdots ,T_{N}\right\}$ can be constructed based on the motif distribution of every node. We further define the directed network node dispersion (DNND) to measure the connectivity heterogeneity between nodes [22]. A larger DNND indicates greater heterogeneity in the connectivity of nodes within the network, while a smaller DNND suggests a more uniform distribution of node connections. And DNND is given by the following formula:(1)$$DNND\left(G\right)=\frac{\zeta (T_{1},T_{2},\cdots ,T_{N})}{\ln\left(N+1\right)},$$
where $\zeta (T_{1},T_{2},\cdots ,T_{N})$ is the Jensen–Shannon divergence of the *N* motif distributions and is given by
(2)$$\zeta \left(T_{1},T_{2},\cdots ,T_{N}\right)=\frac{1}{N}\sum_{i,j}t_{i}\left(j\right)\ln\left(\frac{t_{i}\left(j\right)}{\mu_{j}}\right),$$
where $\mu_{j}$ represents the average value of *N* motif distributions, and the specific calculation is as follows:(3)$$\mu_{j}=\frac{\sum_{i=1}^{N}t_{i}\left(j\right)}{N}$$

Given two directed networks $G_{1}\left(V_{1},E_{1}\right)$ and $G_{2}\left(V_{2},E_{2}\right)$, the structural dissimilarity between them can be calculated based on their motif distribution matrices $T_{1}$ and $T_{2}$. We use $D_{m}\left(G_{1},G_{2}\right)$ to represent the dissimilarity between $G_{1}$ and $G_{2}$, and thus,
(4)$$D_{m}\left(G_{1},G_{2}\right)=\varphi \sqrt{\frac{\zeta (\mu^{G_{1}},\mu^{G_{2}})}{\ln2}}+\left(1-\varphi \right)\left|\sqrt{DNND(G_{1})}-\sqrt{DNND(G_{2})}\right|,$$
where
(5)$$\zeta \left(\mu^{G_{1}},\mu^{G_{2}}\right)=\frac{1}{2}\sum_{j=1}^{35}\mu_{j}^{G_{1}}\ln\left(\frac{\mu_{j}^{G_{1}}}{\mu_{j}^{G_{1}}+\mu_{j}^{G_{2}}}\right)+\frac{1}{2}\sum_{j=1}^{35}\mu_{j}^{G_{2}}\ln\left(\frac{\mu_{j}^{G_{2}}}{\mu_{j}^{G_{1}}+\mu_{j}^{G_{2}}}\right)$$The dissimilarity $D_{m}$ comprises two terms, and we use the parameter φ(0≤φ≤1) to adjust their weights. The first term illustrates the difference between the average motif distributions, that is, $\mu^{G_{1}}=\left(\mu_{1}^{G_{1}},\mu_{2}^{G_{1}},\cdots ,\mu_{N_{1}}^{G_{1}}\right)$ and $\mu^{G_{2}}=\left(\mu_{1}^{G_{2}},\mu_{2}^{G_{2}},\cdots ,\mu_{N_{2}}^{G_{2}}\right)$, and predominantly signifies the global distinctions between the two networks. The second term mainly describes the difference between the *DNND*s of the two networks, indicating the local difference between them. A lower value of $D_{m}$ indicates a higher network similarity and vice versa.

## 3. Baselines and Datasets

### 3.1. Baselines

**Portrait-based directed network comparison method** [23]: For a directed network *G*, we construct a portrait matrix *B* based on the distance between nodes. Each element $B_{l,k}$ represents the number of nodes that have *k* nodes at distance *l*, where 0≤l≤d, 0≤k≤N−1, and *d* represents the diameter of *G*. We note that we utilize the shortest directed path length to calculate the distance between nodes. In addition, *B* is independent of the ordering and labeling of the nodes. Based on $B_{l,k}$, we can derive the probability that a randomly selected node has *k* nodes at a distance of *l* and is given by
(6)$$Q_{l,k}=\frac{1}{N}B_{l,k}$$

For two directed networks, $G_{1}$ and $G_{2}$, the probability distributions $Q_{1}$ and $Q_{2}$ are employed to interpret the rows of the network portraits for each of them. The similarity between $G_{1}$ and $G_{2}$ is represented by $D_{p}\left(G_{1},G_{2}\right)$ and is defined as follows:(7)$$D_{p}\left(G_{1},G_{2}\right)=\frac{1}{2}KL\left(Q_{1}\|M\right)+\frac{1}{2}KL\left(Q_{2}\|M\right),$$
where $M=\frac{1}{2}\left(Q_{1}+Q_{2}\right)$, and $KL\left(*\|*\right)$ represents the Kullback–Liebler divergence between two distributions.

**DeltaCon-based directed network comparison method** [24]: DeltaCon considers the similarity between two networks by quantifying the difference between r-step paths other than the edges. Given a directed and unweighted network *G* and its adjacency matrix *A*, the r-step paths are encoded in the similarity matrix $S={\left[I+\varepsilon^{2}D-\varepsilon A\right]}^{-1}$, where *D* and *I* are diagonal matrices with diagonal elements equal to the node degree and 1, respectively, and $\varepsilon =1/\left(1+\max\left(D_{ii}\right)\right)\left(i=1,\cdots ,N\right)$. We assume that the similarity matrices for two directed and unweighted networks $G_{1}$ and $G_{2}$ are denoted by *S* and $S^{\prime }$, and the dissimilarity $D_{d}$ between them is given by the following equation:(8)$$D_{d}\left(G_{1},G_{2}\right)={\left\{\sum_{i,j=1}^{N}{\left(\sqrt{S_{ij}}-\sqrt{S_{ij}^{\prime }}\right)}^{2}\right\}}^{\frac{1}{2}}$$

**Closeness-based directed network comparison method:** Centrality measures, such as degree, betweenness, and closeness, were used to compare networks [27]. However, in this part of the experiments, we found that closeness centrality surpasses other centrality methods in network comparison. Therefore, we omit the other centrality measures and only use closeness for directed network comparison. Closeness centrality measures the importance of a node within a network by evaluating the proximity of its connections to other nodes. The closeness centrality of a node is defined by
(9)$$c_{i}=\frac{1}{\sum_{i\neq j}d_{ij}},$$
where $d_{ij}$ represents the directed shortest path length from node $v_{i}$ to node $v_{j}$. For two directed networks $G_{1}$ and $G_{2}$, we assume that the closeness centrality vectors for them are given by $c={\left(c_{1},c_{2},\cdots ,c_{N}\right)}^{T}$ and $c^{\prime }={\left(c_{1}^{\prime },c_{2}^{\prime },\cdots ,c_{N}^{\prime }\right)}^{T}$. Therefore, the dissimilarity between $G_{1}$ and $G_{2}$ based on closeness centrality is given as follows:(10)$$D_{c}\left(G_{1},G_{2}\right)=\sum_{i=1}^{N}\left|c_{i}-c_{i}^{\prime }\right|$$

### 3.2. Description of Directed Network Datasets

To evaluate the performance of our proposed methods and the state-of-the-art baselines, we selected six real-world directed networks from diverse domains, including biological networks, transportation networks, and social networks. The descriptions of each of the datasets are as follows:

**Mac** [36] describes the interactions between adult female Japanese macaques and is about the dominance behavior between them. Each node denotes a macaque, and a directed edge from node $v_{i}$ to $v_{j}$ indicates the dominance of $v_{i}$ over $v_{j}$.

**Caenorhabditis elegans (Elegans)** [37] is a neural network of Caenorhabditis elegans. It uses directed edges to represent neural connections among neurons in the nervous system of Caenorhabditis elegans.

**Physicians** [38] is a directed network that describes the spread of innovation among physicians. A directed edge $\left(v_{i},v_{j}\right)$ between two physicians $v_{i}$ and $v_{j}$ implies that $v_{i}$ would turn to $v_{j}$ if he or she needs suggestions or is interested in a discussion.

**Email-Eu-core (Email)** [39] is an email network that captures email interactions between institution members in a large European research institution. A directed edge between two staff members $v_{i}$ and $v_{j}$ means that staff member $v_{i}$ has sent an email to staff member $v_{j}$.

**US airport** [40] illustrates the flight connections between US airports. A directed edge $\left(v_{i},v_{j}\right)$ between two airports $v_{i}$ and $v_{j}$ illustrates that there is at least a flight from airport $v_{i}$ to $v_{j}$.

**Chess** [40] is a network that characterizes the interaction between players in an international chess game within a month. A directed edge is formed from a white player to a black player in this network.

Table 1 shows the basic properties of the directed networks mentioned above, including the number of nodes $\left(N\right)$, the number of edges $\left(M\right)$, the average degree $\left(Ad\right)$, the average shortest path length $\left(Avl\right)$, and the network diameter $\left(d\right)$.

## 4. Experimental Results

### 4.1. The Dissimilarity between a Real Network and Its Null Models

The null model is widely used as a tool for the comparison of network topology [41] and retains specific network properties, such as the degree distribution or clustering coefficient, via the random reshuffling of network connections. In this section, we propose three null models for directed networks to gradually change the network topology and use our comparison method to compare each directed network and its null models.

We extend the dk-series null models that were originally proposed for undirected networks to directed networks [42], which retain the degree distributions, correlations, and clustering of a real directed network to some extent. Concretely, the models are illustrated as follows: Dk1.0 preserves the out-degree and in-degree of a node by randomly rewiring each directed edge. Therefore, the degree sequence of the original network is preserved in the reshuffling process. Dk2.0 reshuffles every edge in the network while maintaining the out-degree, in-degree, and joint degree distribution of the original network. Dk2.5 rewires every edge by preserving the distribution of the degree-dependent clustering coefficient. We note that the newly formed directed edges should never have existed in the original network.

We show examples of how to generate the null models in Figure 2a–c, in which the blue dashed lines indicate the newly connected edges. The left panel shows the original network, and the right panel shows the network after the rewiring process in each of the figures. Figure 2a shows an instance for Dk1.0. Specifically, we disconnect the edges $\left(v_{1},v_{2}\right)$ and $\left(v_{3},v_{4}\right)$ and form new edges, i.e., $(v_{1}$, $v_{4})$ and $(v_{3}$, $v_{2})$. Therefore, the in-degree and out-degree of each node are preserved in this process. Figure 2b demonstrates the generation of a random network through Dk2.0, which is more strict than Dk1.0. For example, if we disconnect the directed edge between $v_{1}$ and $v_{2}$, that is, $\left(v_{1},v_{2}\right)$, we need to find a node that has the same in-degree and out-degree as $v_{2}$, and the appropriate node is $v_{4}$. Accordingly, we connect $v_{1}$ and $v_{4}$ and form a new directed edge $\left(v_{1},v_{4}\right)$. Therefore, Dk2.0 maintains the degree sequence and the joint degree distribution of a network. In Figure 2c, the degree (sum of in-degree and out-degree) and clustering coefficient for each node are {2,3,3,3,3,1,1,1,1} and {1/2,1/6,0,0,1/6,0,0,0,0}, respectively. Therefore, the average clustering coefficients for nodes that have degrees of {1,2,3} are {0,1/2,1/12}, respectively, which are also called degree-dependent clustering coefficients. We disconnect the directed edges $\left(v_{1},v_{2}\right)$ and $\left(v_{4},v_{3}\right)$ and form new directed edges as $\left(v_{1},v_{3}\right)$ and $\left(v_{4},v_{2}\right)$. In the rewired network, the degree-dependent clustering coefficient distribution is the same as that in the original network.

A lower value of *k* implies a greater disruption of the original network structure. In Figure 3, we use the four directed network comparison methods mentioned above to quantify the dissimilarity between each of the directed networks and its three null models. Figure 3a–d represent the comparison results via $D_{m}$, $D_{p}$, $D_{d}$, and $D_{c}$, respectively. The experimental results for six networks suggest that as *k* increases, the similarity between the original network and its null models gradually increases for our proposed method. The dissimilarity observed in our approach aligns with the generation of null models, providing further confirmation of the effectiveness and stability of our model in comparing directed networks from different domains. For the baseline models, $D_{d}$ shows a similar trend to our method. However, $D_{p}$ and $D_{c}$ show bad performance in networks such as Email and Physicians, respectively.

### 4.2. The Comparison of the Directed Network and Its Perturbed Network

In this section, we report perturbation experiments performed on the edges of six real directed networks to further assess the stability and applicability of the motif-based comparison method. Specifically, for each given network, we randomly add or remove edges with a certain proportion, *f*, where the range of *f* is $\left[-0.9,0.9\right]$. The positive value of *f* indicates that we randomly add an |f| fraction of directed edges to the network, and the negative value of *f* means that we randomly remove an |f| fraction of directed edges. We compare the original network with the perturbed network by adding or removing edges using different network comparison methods, as shown in Figure 4. The four comparison methods ($D_{m}$, $D_{p}$, $D_{d}$, and $D_{c}$) show similar trends; that is, the increase in |f| will make the perturbed network have a greater difference from the original network, which is consistent with intuition. This conclusion is especially significant when *f* is negative, where our method, as well as the baselines, can significantly distinguish between the original network and the ones after perturbation. However, the motif-based comparison method is much better than the rest of the baselines for positive values of *f*. The curves of the other three baselines for f>0 are flatter than those of our method. Taking the Mac network as an example (Figure 4a), the values of $D_{p}$ range from 0.07 to 0.13 for f∈[0,1], and the values of $D_{p}$ are the same for f=0.1 and f=0.2, which is unreasonable. $D_{d}$ and $D_{c}$ also show insignificant dissimilarities between networks in Figure 4a–f. The baseline methods, such as $D_{p}$ and $D_{c}$, are based on the distance between nodes, and $D_{d}$ considers the r-step paths of a network for network comparison. However, they do not consider the higher-order organization of a network, i.e., at the level of subgraphs, and thus may result in poor performance in network comparison. For example, the distance between nodes is obtained through pairwise interactions (i.e., edges) between nodes. In addition, the r-step paths are also constructed using edges.

### 4.3. Parameter Sensitivity Analysis

The motif-based directed network comparison method involves a parameter, denoted by φ, that determines how much importance is given to the global or local differences between two networks, with a larger value of φ indicting that we consider more of global difference and vice versa. Therefore, we performed a parameter analysis for φ in the six real-world directed networks via the comparison of the original network and its perturbed networks. The results are given in Figure 5, in which we use curves with different colors to indicate different chosen values of φ(φ∈{0.1,0.3,0.5,0.7,0.9}). The figure displays curves that exhibit a similar trend for different values of φ, and there is small deviation among the curves when f<0. However, the network dissimilarity for different *f* is more significant for φ=0.5 in most networks (except Physicians and Email), which means that we need to consider the global or local differences between networks for comparison. Therefore, we used φ=0.5 in the above analysis.

## 5. Conclusions

In this paper, we introduce a comparison method $D_{m}$ that utilizes network motifs to assess similarities between directed networks. The method, which considers both local and global differences between two directed networks as well as higher-order information, is based on node motif distributions and employs the Jensen–Shannon divergence. In detail, we use motifs with sizes up to 4 that are listed in Figure 1 to compute the motif distribution of nodes in a directed network. Based on the Jensen–Shannon divergence and motif distributions of nodes, we define the dispersion of directed network nodes (DNND) to quantify the heterogeneity of connectivity between nodes. Lastly, for two given directed networks, the similarity between them is further defined by the combination of the DNND metrics and the average motif distributions. Our method aims to better understand the internal connection patterns of the network nodes by capturing essential subgraph structures. To show the effectiveness of our method, we compare a directed network with its null models, which gradually change the structure of the original network. In addition, we further compare our method with the baselines to characterize the similarity between an original network and its perturbed networks. The results show that our method outperforms these baseline methods across networks from different domains.

Motifs have been widely used to address a range of tasks. In our analysis, we take into account the directionality of edges by utilizing directed motifs to compare directed networks. We limit our analysis to motifs with sizes up to 4 due to the high computational expenses involved. It is worthwhile to investigate the impact on network comparison performance when analyzing motifs with varying numbers of nodes. Although considering larger motifs could potentially enhance the effectiveness of our approach, it may pose scalability challenges when dealing with large networks containing millions of nodes. Given the success of motifs in network comparison, we believe that developing efficient algorithms for computing motifs could be a promising avenue for research. This has the potential not only to enhance network comparison but also to improve other network tasks, such as community detection, node classification, influence maximization, and more.

## Figures and Tables

**Figure 1 entropy-26-00128-f001:**
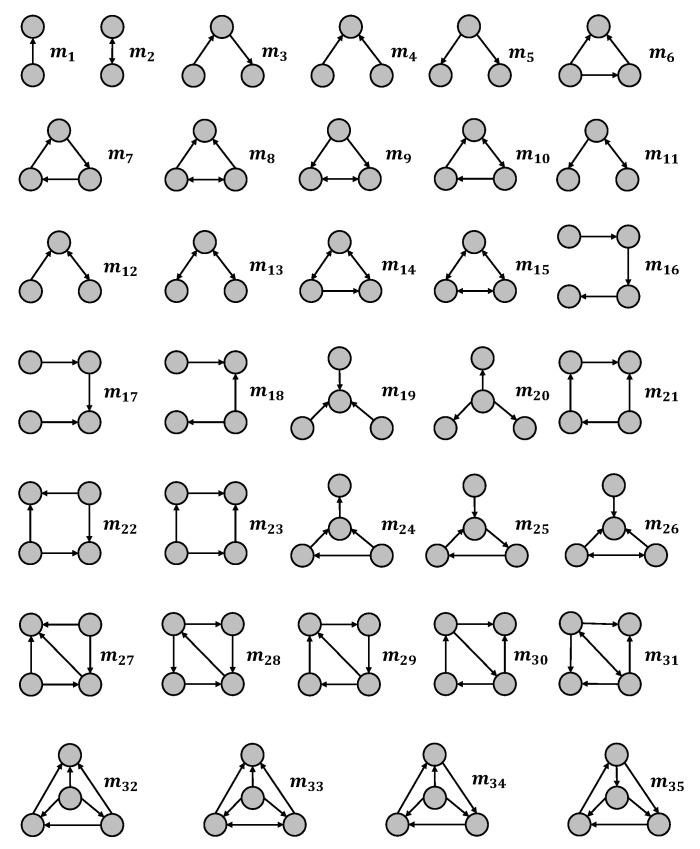
Motifs formed by 2 to 4 nodes in directed networks. All the motifs are labeled from $m_{1}$ to $m_{35}$.

**Figure 2 entropy-26-00128-f002:**
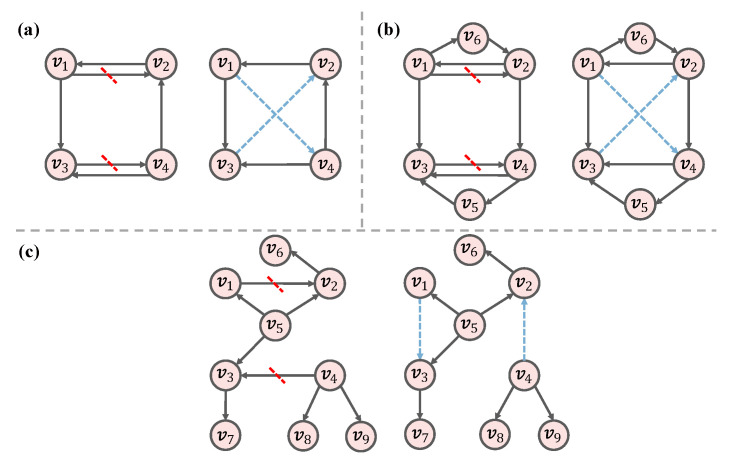
Toy examples of three dk-series null models: (**a**) Dk1.0; (**b**) Dk2.0; (**c**) Dk2.5. The blue dashed lines indicate the newly connected edges. In (**a**–**c**), the left panel shows the original network, and the right panel shows the rewired network.

**Figure 3 entropy-26-00128-f003:**
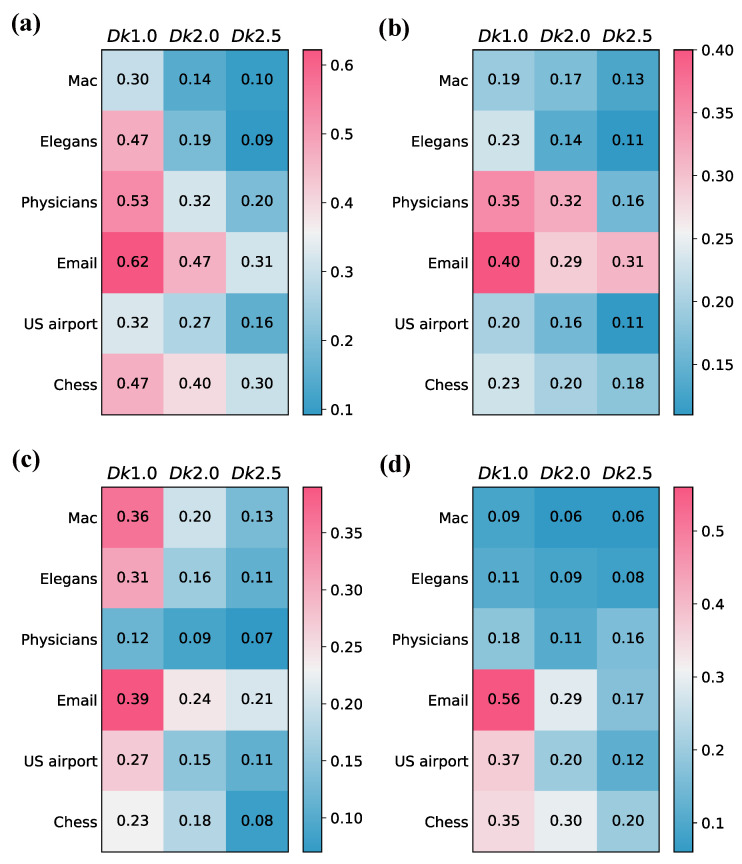
Comparison between real directed networks and their null models via motif-based directed network comparison method and baseline methods. The null models are Dk1.0, Dk2.0, and Dk2.5. With the increase in *k*, more topological properties of the original network will be preserved. We show results for different methods: (**a**) $D_{m}$; (**b**) $D_{p}$; (**c**) $D_{d}$; (**d**) $D_{c}$. Smaller values in the heatmap indicate a higher similarity, and vice versa. The results are the average of 100 realizations.

**Figure 4 entropy-26-00128-f004:**
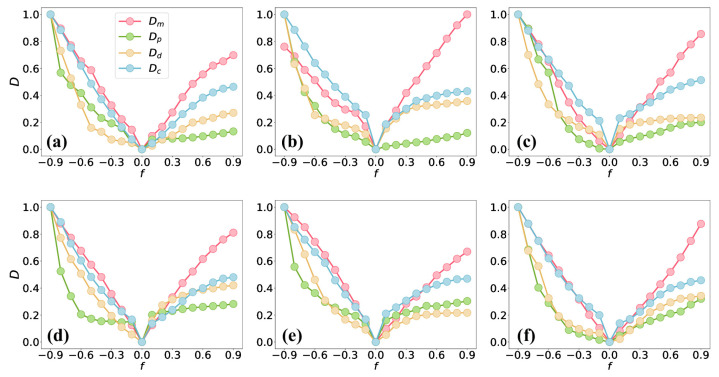
Similarity between a real directed network and a perturbed network generated by randomly adding or deleting edges, where positive values of *f* indicate that we randomly add an *f* fraction of edges, and vice versa. We show the results for the following networks: (**a**) Mac; (**b**) Elegans; (**c**) Physicians; (**d**) Email; (**e**) US airport; (**f**) Chess. The parameter φ of $D_{m}$ is set to 0.5. Each point in the figure is averaged over 100 realizations.

**Figure 5 entropy-26-00128-f005:**
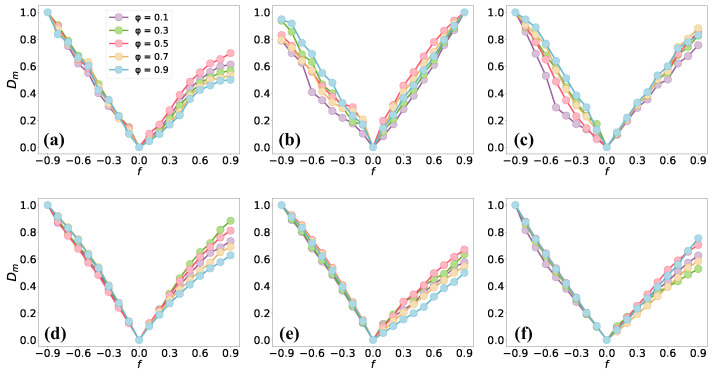
Parameter analysis for motif-based directed network comparison. We compare the real network with its perturbed network via edge addition or deletion. Different curves show different chosen values of φ, which is the only parameter in our method, $\varphi \in \left\{0.1,0.3,0.5,0.7,0.9\right\}$. Positive values of *f* indicate a random edge addition, and vice versa. We show the results for the following networks: (**a**) Mac; (**b**) Elegans; (**c**) Physicians; (**d**) Email; (**e**) US airport; (**f**) Chess. All results are averaged over 100 realizations.

**Table 1 entropy-26-00128-t001:** Basic properties of real directed networks, where *N*, *M*, Ad, Avl, and *d* represent the number of nodes, the number of edges, the average degree, the average shortest path length, and the network diameter, respectively.

Networks	*N*	*M*	*Ad*	*Avl*	*d*
Mac	62	1187	38.29	1.38	2
Elegans	237	4296	28.92	2.47	5
Physicians	241	1098	9.11	2.58	4
Email	1005	25,571	50.84	2.94	7
US airport	1574	28,236	35.87	3.13	8
Chess	7301	65,053	17.82	3.92	13

## Data Availability

Data will be available on request.

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
