# Peer review of "Directed Network Comparison Using Motifs"

_entropy, 2024, doi:10.3390/e26020128_

Round 1
Reviewer 1 Report
Comments and Suggestions for Authors
The manuscript proposes a novel motif-based network comparison method that can capture the local and global differences between networks simultaneously. The experimental results validate the robustness and effectiveness of the proposed method. However, the article presents the following issues:
1. In the introduction section, the authors mentioned that previous research has already compared directed networks based on graphlets. How does the method proposed in this paper differ from these graphlet-based methods?
2. In section 4.1, the similarities between real directed networks and their null models are compared using the method proposed in this manuscript. Here, it is necessary to add some comparison results between these networks and their null models via the three baseline methods selected in this manuscript.
3. The experimental results in Figure 3 should be averaged. Please indicate in the caption how many times each result is averaged.
4. The citations for some references in certain sections are incorrect, such as references [38] and [39]. It is recommended to carefully review the citation references throughout the entire document.
Comments on the Quality of English Language
There are some punctuation and spelling errors in the manuscript. For example, in the conclusions section, there is a missing period between the last sentence and the preceding one, and in the last sentence, "original" should be corrected to "original". Similar errors are present throughout the entire manuscript. It is recommended to conduct a comprehensive review and make the necessary corrections.
Reviewer 2 Report
Comments and Suggestions for Authors
The authors have proposed a comparison method between two directed networks, for which the variations of distributions of characteristic motifs are utilized to measure the dissimilarity, and numerically shown that the proposed method by an entropy-like measure outperforms the conventional baseline methods for almost all of tested network data in comparing with random null models. These results are interesting and will give the effectiveness for several applications in network science. However, the following points can be improved.
1. Introduction
In line 8 from the top on page 2, although the authors describe "most of them have not considered higher-order structure of a network", the definition of higher-order structure is not explained. I wonder that motifs are not higher-order but local subgraph's ones. Thus, only the above description seems to be weak for a reason why motifs are considered. The authors should clearly explain the motivation. The word "higher-order structure" may cause confusing and unsuitable, therefore in stead of using it, "interactions of nodes until four (with not necessarily direct connections)" is recommended, for example.
2.2 Motif-based Directed Network Comparison Method
For Eq.(1) in page 3, why is the denominator $\ln (N+1)$ necessary ? On page 4, why is the Jensen-Shannon (J-K) divergence of Eq.(2) applied ? The authors should explain these reasons and that the K-J divergence is usually applied for what purposes with a related reference to it. In addition, there is no definition of $\zeta(\mu_{G_{1}}, \mu_{G_{2}})$ in the 1st term of left-hand side of Eq.(4).
4.1 The dissimilarity between a real network and its null models
Around the bottom on page 7, it is better to note that the similarity increases as higher $k$ in dk-series null models, because such null models remain much more topological properties including not only degrees but also degree-degree correlations and other higher correlated structures of more than two connected nodes to the original network.
4.2 The comparison of directed network and its perturbed network
Around the bottom on the 1st paragraph on page 8, the authors should explain why and how is the higher-order structure ignored in the baseline models ? I think that the distances between nodes or the shortest paths used in the baseline models are considered as a kind of higher-order structure related to more than two nodes. In addition, for Figure 4, the authors should describe that the conventional $D_{c}$ is higher than proposed $D_{m}$ as shown by blue and magenta curves when $f < 0$ in (b)(c) even as exceptions.
5. Conclusion
This paper handles the motifs until four nodes for reducing computation. However, smaller or larger motifs of three or more than four nodes can be considered. At least, it should be mentioned to comparatively analyze the performance for these cases as a future study in the last section.
Comments on the Quality of English Language
2.1 The definition of motifs in a directed network
... are the node and edge set, respectively.
→ ... are the sets of nodes and edges, respectively.
The number of nodes and the number of edges are ...
→ The numbers of nodes and edges are ...
